# Machine Learning Predictive Model as Clinical Decision Support System in Orthodontic Treatment Planning

**DOI:** 10.3390/dj11010001

**Published:** 2022-12-20

**Authors:** Jahnavi Prasad, Dharma R. Mallikarjunaiah, Akshai Shetty, Narayan Gandedkar, Amarnath B. Chikkamuniswamy, Prashanth C. Shivashankar

**Affiliations:** 1Department of Orthodontics and Dentofacial Orthopedics, DAPM R V Dental College and Hospital, Bengaluru 560078, Karnataka, India; 2Senior Lecturer—Orthodontics, Discipline of Orthodontics & Pediatric Dentistry, School of Dentistry, University of Sydney, Sydney, NSW 2006, Australia

**Keywords:** machine learning, orthodontic treatment planning, clinical decision support system

## Abstract

Diagnosis and treatment planning forms the crux of orthodontics, which orthodontists gain with years of expertise. Machine Learning (ML), having the ability to learn by pattern recognition, can gain this expertise in a very short duration, ensuring reduced error, inter–intra clinician variability and good accuracy. Thus, the aim of this study was to construct an ML predictive model to predict a broader outline of the orthodontic diagnosis and treatment plan. The sample consisted of 700 case records of orthodontically treated patients in the past ten years. The data were split into a training and a test set. There were 33 input variables and 11 output variables. Four ML predictive model layers with seven algorithms were created. The test set was used to check the efficacy of the ML-predicted treatment plan and compared with that of the decision made by the expert orthodontists. The model showed an overall average accuracy of 84%, with the Decision Tree, Random Forest and XGB classifier algorithms showing the highest accuracy ranging from 87–93%. Yet in their infancy stages, Machine Learning models could become a valuable Clinical Decision Support System in orthodontic diagnosis and treatment planning in the future.

## 1. Introduction

In recent years, with the flourishing of studies and technological evolution, artificial intelligence (AI) is going through revolutionary progress in the field of medicine. The development of medical artificial intelligence [1] has been identified with the development of Clinical Decision Support Systems (CDSSs) [2,3] which are interactive computer programs designed and intended to help the clinician to develop a diagnosis and determine appropriate treatment options and prognosis.

Machine Learning (ML) [4,5], a branch of AI, facilitates machines and computer systems to process, analyze and interpret data to aid in providing solutions for real-world challenges such as software and website designing, online data storage, sharing and protection, national security and defense and health care management (patient appointment management, treatment predictions, robotic surgeries, 3D printing, etc.) The algorithms have the ability to learn and execute actions on their own based on the type of adequate data provided. The algorithm is set up so that machines can anticipate results dependent on previous events. With the provided information, Machine Learning algorithms and strategies help in training a model to foresee and conform to future outcomes.

In the field of dentistry, various AI–ML-integrated programs have been developed to assist dental specialists in their clinical practice. Some of them include diagnosis of proximal dental caries [6], AI in dental surgery [7], categorize pre-malignant and malignant oral smears [8] apical foramen localization [9], oral cancer prognosis [10], etc.

Diagnosis and treatment planning forms the essence of orthodontics. Many studies have been carried out in orthodontics that have made use of AI and ML to predict various aspects of treatment planning, most of which deal with cephalometric landmark identification [11,12,13,14,15,16,17,18,19] and extraction–non-extraction decisions [20,21,22,23,24,25]. However, there are no studies yet to predict a broader outline of the diagnosis of a patient’s malocclusion, and the mode of treatment as comprehensive and deliberate evaluation of numerous factors makes diagnosis and treatment planning a complex process. Moreover, diagnosis depends on the subjective judgement of the orthodontists, with various philosophies and theories of orthodontics posing a challenge to standardize a particular treatment, thus leading to inter- and intra-clinician incongruities and also differences in treatment plans between experienced and less experienced orthodontists.

Orthodontists acquire keen clinical expertise with many years of practice and experience, along with the support of essential and supplemental diagnostic aids. Machine Learning models, which have the ability to learn by pattern recognition and algorithms using data fed into a computer, can gain this expertise using the algorithm in a very short timeframe. They use a training dataset of large data collected from many expert orthodontists. By this method of learning, there is diminished odds of error, missing out on data, inter- and intra-clinician discrepancies and enhancement of the accuracy in prediction of diagnosis and treatment planning [21].

Therefore, the aim of this study was to evaluate a Machine Learning predictive model for orthodontic treatment plan prediction and to correlate the treatment plan prediction of the model to the treatment plan of experienced orthodontists.

## 2. Materials and Methods

A total of 700 case records satisfied the inclusion criteria, which included (i) case record files of the patients containing clinical examination, cephalometric data and the treatment plan formulated by orthodontists and (ii) patient age groups between 10–30 years who have undergone orthodontic treatment including growth modulation, camouflage and jaw surgery modes of treatment in the past ten years. They were procured from the Department of Orthodontics and Dentofacial Orthopedics. The exclusion criteria for the study were (i) patients with significant systemic medical history and conditions, (ii) patients with developmental disturbances, mutilated dentition, syndromes, craniofacial defects and gross facial asymmetries and (iii) patients with a previous history of orthodontic treatment.

The sample was divided into a training set and a test set in a 70:30 ratio. A basic outline of orthodontic diagnosis and treatment planning was designed for model construction and training (Figure 1).

The input layer of variables consisted of parameters that most commonly determine the diagnosis and treatment planning of the patient (Table 1). All the layers consisted of 33 parameters. The output layer of 11 variables was comprised of the diagnosis and mode of treatment. These input variables were processed to ensure all of them were quantified by converting them into numerical values (Table 2—Data Dictionary) before being used for the model training. The data entry was conducted in four Microsoft Excel spreadsheets, one for each model. The model training was performed with the training set.

The quantified data, including the treatment mode (determined by 10–15 expert orthodontists with nearly 10–15 years of experience), were fed into the model. Through pattern recognition, the first model was trained to diagnose the skeletal jaw bases I, II and III. Once the jaw bases were diagnosed, the second model predicted the two broad treatment modes for skeletal jaw base I—whether to extract (Extraction) or not (non-Extraction of first and/or second bicuspids). The third and the fourth models predicted the treatment modes of skeletal class II and skeletal class III—whether the patient requires growth modulation, camouflage or jaw surgery treatment (Figure 2).

During training of the model, the test set was not accessible to the model set. The seven most suitable algorithms were used to construct the ML model and the data were run with all the algorithms (Table 3).

Once the model was trained, the test set with only the input parameters without the output (treatment) was fed. The final prediction of the ML predictive model was compared and correlated with the treatment plan formulated by the orthodontists for the same test set of cases (Figure 3). Thus, the accuracy and efficiency of the model was determined.

## 3. Results

According to the results, the Machine Learning algorithmic predictive models showed accuracy F1 values of 84.93% for layer 1 (Table 4, Figure 4), 82.22% for layer 2 (Table 5, Figure 5), 81.51% for layer 3 (Table 6, Figure 6) and 87.08% for layer 4 (Table 7, Figure 7), with eXtreme Gradient Boosting (XGB), Random Forest (RF) and Decision Tree (DT) showing the highest accuracy in the prediction.

The overall model (Table 8, Figure 8) and test set (Table 9, Figure 9) showed an average accuracy of 84%, which means the model’s prediction of the treatment plan was the same as the treatment plan decided by the orthodontists in 84% of total cases.

The correlation between the input parameters was also determined with a heat map (Figure 10) of the correlation and regression analysis matrix. Furthermore, in this study, the relative contribution of the individual parameters or factors to the treatment plan was determined, and the top ten parameters were ranked in ascending order (Figure 11).

## 4. Discussion

The present study aimed to explore the possibilities of the application of an AI–ML predictive model as a Clinical Decision Support System (CDSS) in orthodontic treatment planning.

Park et al. [16,17] compared two AI machine learning algorithms for automated identification of cephalometric landmarks and concluded that the AI accurately identified the landmarks with approximately 50% accuracy. Kök et al. [26] used AI algorithms for the determination of growth by Cervical Vertebrae Maturation Stage (CVMI) stages in orthodontics using cephalometric radiographs. The study used algorithms similar to our study—Random Forest Classifier, Logistic Regression, Decision Tree Classifier, K-Neighbors Classifier, Artificial Neural Network (ANN), Linear Support Vector Machine (SVM) and Naïve Bayes Classifier with Decision Tree showed the highest accuracy.

Yu et al. [27] used a deep learning multimodal Convoluted Neural Network (CNN) model for skeletal classification using cephalometry in orthodontic diagnosis and treatment planning. The model showed high performance in classifying skeletal jaw bases. Although deep learning algorithms have been shown to improve performance when applied to cephalometric analysis, many of these studies focus on detecting cephalometric landmarks [11,12,13,14,15,16,17,18,19,28,29].

Compared with the above studies, which made use of cephalometric data alone, in this study, we used a trained model which included clinical, photographic and cephalometric data for its prediction (Figure 2). Moreover, in comparison with that of Kök et al. [26], three of the algorithms, namely Decision Tree, Random Forest and XGB classifier, showed high accuracy, from 87–90% individually in treatment plan prediction.

Jung et al. [20] used ANN for diagnosis of extractions and extraction patterns. In addition to cephalometric measurements, six indexes—maxillary arch length discrepancy index, mandibular arch length discrepancy index, molar key index, large overjet index, protrusion index and chief complaint index for protrusion—were included in the input data [20]. The study was conducted with 156 subjects with 80% accuracy. Peilin Li et al. [21] used a multilayer perceptron ANN for the determination of extraction–non-extraction, extraction patterns and anchorage type in 302 subjects with 82% accuracy. Xi et al. [30] constructed a decision-making ES using ANN to determine the necessity of orthodontic extractions in patients between 11–15 years old.

The above studies have been conducted majorly to determine the need for extractions and their patterns in orthodontics, with approximately 150–300 subjects using ANN. However, our study used and compared 6 different algorithms (Table 3) for a much larger sample size of 700 to improve the performance of the model and dealt not only with extraction decisions but also classifying skeletal jaw bases and different modes of treatment, such as growth modulation, camouflage and jaw surgery treatments, which has not been done so far. Compared with other studies, the ML model used in our study achieved an improved accuracy of 84% (Table 8), with individual layer accuracy going up to 87%.

Furthermore, not only a relative contribution of factors was determined, but also the inter-factor correlation was deduced with heat maps and charts.

According to the heat map analysis (Figure 10) having the same parameters on the X and Y axes, the highest correlation (varying shades of green indicating the strength) was been found between:Age and CVMI stages;Lip competency and lip strain;Profile, beta angle and mandibular dimensions;SNA angle, N perpendicular Pt. A and maxillary dimensions;Upper and lower incisor inclinations to interincisal angle;Overjet and overbite;Lower incisor inclination and IMPA;ANB angle and overjet.

The least correlation (varying shades of red indicating the strength) was found between:ANB angle and beta angle;Overjet and beta angle;Overbite and beta angle;Profile and overjet;Interincisal angle to IMPA;Profile and ANB angle.

The feature importance graph (Figure 11), with the percentage of correlations (highest 100% and least 0%) in the X axis and parameters in the Y axis, revealed that Beta angle, ANB angle, wits appraisal, clinical profile, overjet and overbite and maxillary and mandibular dimensions contribute the highest in the orthodontic treatment plan prediction.

Diagnosis and treatment planning are extremely important in orthodontics [31]. This is because many treatments are irreversible or cause irreversible adverse effects, such as apical root resorption, gingival recession and dental caries. Because of this irreversibility, the recall and F1 scores of the ML model is of more importance for reliability in its prediction. Therefore, the system output showed the highest F1 accuracy value in treatment prediction (Table 4, Table 5, Table 6, Table 7, Table 8 and Table 9). This supports efficient implementation of the model for clinical practice.

Treatment plans that do not consider the discrepancies of skeletal components and its severity are inconsistent. To assess the model’s ability to classify a skeletal discrepancy, layer 1 was constructed to identify the jaw bases as class I, II and III (Figure 4). Following this, layer 2 was constructed to predict two major modes of treatment—extraction and non-extraction for normal jaw base I [32,33,34,35] (Figure 5). However, due to increased complexity and a limited dataset for training, various other modes of treatment such as expansion, distalization, etc., were not considered.

Three major treatment options for skeletal jaw bases II (retrusive profile) and III (protrusive profile) are growth modulation, camouflage and jaw surgery [36,37,38,39,40]. Layer 3 and 4 were constructed for jaw bases II and III, respectively, with data which contributed towards the treatments (Figure 6 and Figure 7). Additional in-depth options such as appliance selection, camouflage patterns and jaw surgery procedures were not considered, again due to the limited dataset and the complexity of coding.

Medical data are often too complex for detailed AI analysis. Thus, despite the high performance of the ML model in the present study, the main limitation was the amount of data (e.g., skeletal class II cases are more prevalent, class III jaw surgery cases are rare due to patient preferences, crowding cases are more prevalent, demanding extractions, etc.), so the model was trained with data that could have been affected by selection bias. To avoid the bias, an almost equal number of cases were taken for every layer.

Orthodontic diagnosis and treatment planning [41,42,43,44,45,46] are highly subjective and opinion based, which vary with the knowledge and expertise [47] of the orthodontists, causing inter- and intra-clinician errors. They are also intricate and multifactorial, with the consideration of innumerable factors such as facial appearance, skeletal relation, patients’ general dental condition, etc. The growth modulation, camouflage and jaw surgery planning [48] require a thorough and careful examination, which is not all about measurements and values. The ML model could not include complex orthodontics cases such as skeletal deformities, uncommon extraction patterns, soft tissue functions, etc.

There are various ways to treat a malocclusion. There is no definite answer on how to treat an orthodontic patient. The purpose of this study was not to find the correct answer. It was to assess whether the ML predictive model could provide a reference or assistance to less experienced practitioners by emulating the dynamics of experienced orthodontists.

The ML model constructed in this study can be further improved with detailed data which are of complex nature using various theories and philosophies of orthodontic diagnosis and treatment planning. It can be integrated with already existing orthodontic software [48,49] and programmed to naturally gauge the information for a comprehensive structured and customized treatment plan for every individual. The field of orthodontics requires more studies and research to explore and analyze the applicability and efficacy of AI and ML. The wide variety of clinical data collected from different clinicians is of great help in model training to improve performance. From an epidemiological point of view, this model is a useful tool for analyzing large amounts of existing data for an elementary classification of the treatment options in order to create large datasets for future retrospective studies.

## 5. Conclusions

Orthodontic diagnosis and treatment planning is undergoing remarkable changes with greater emphasis on soft tissue adaptation and proportions rather than the previous notion of dental occlusion and hard tissue relationships. A comprehensive diagnosis embodies a ‘Problem-Oriented Approach’ comprising appropriate patient history, clinical examination, study model, cephalometric evaluations, etc., which makes it a multifaceted, intricate and highly subjective process.

An elementary attempt was made through this study to predict the diagnosis and treatment plan using an artificial intelligence–Machine Learning model for patients requiring orthodontic treatment, and the efficacy was compared with that of expert orthodontic decisions. Overall, the ML-based AI model showed 84% accuracy in its treatment plan prediction compared with the treatment plan for the same cases decided by expert opinion of orthodontists. It also predicted the relative contribution of individual diagnostic data in the treatment planning decision.

Machine Learning models may not have subtle, expert decision-making ability due to the limited quality of technological expertise and training data, but with rapidly advancing AI innovations and continuous improvement of the diagnostic system by improving the quality and quantity of data, Machine Learning predictive models can be an effective Clinical Decision Support System for orthodontists in the near future and provide orthodontists a diagnostic framework, flexibility and feasibility of different treatment options.

## Figures and Tables

**Figure 1 dentistry-11-00001-f001:**
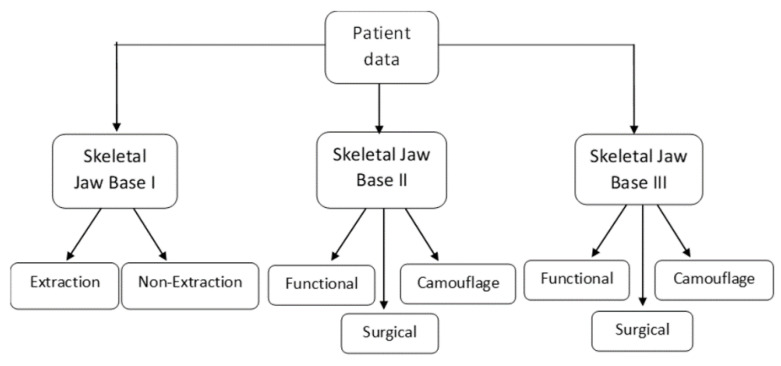
Outline of orthodontic diagnosis and treatment plan.

**Figure 2 dentistry-11-00001-f002:**
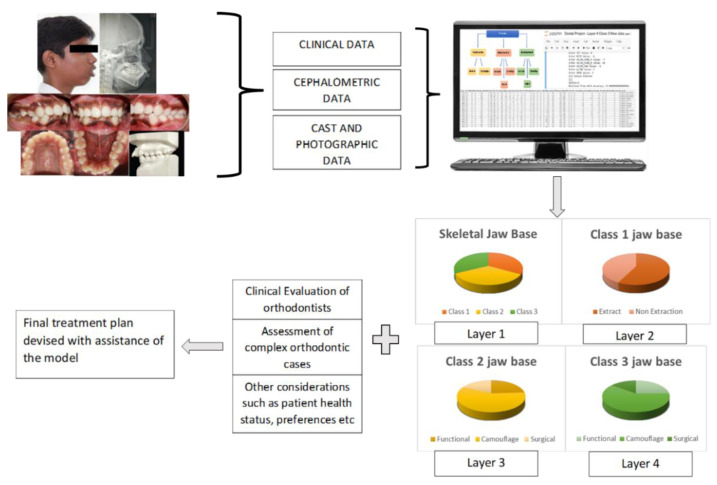
Illustration of computerized, quantified patient data and prediction of all layers. The Machine Learning model along with other considerations can aid in the formulation of the final treatment plan.

**Figure 3 dentistry-11-00001-f003:**
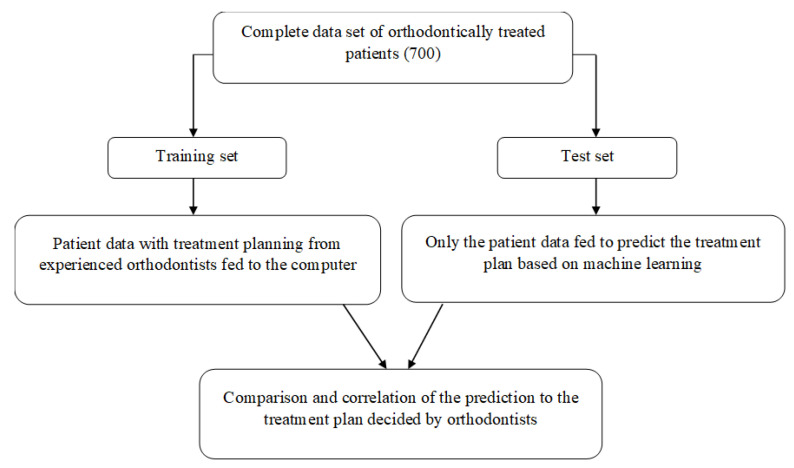
Methodology.

**Figure 4 dentistry-11-00001-f004:**
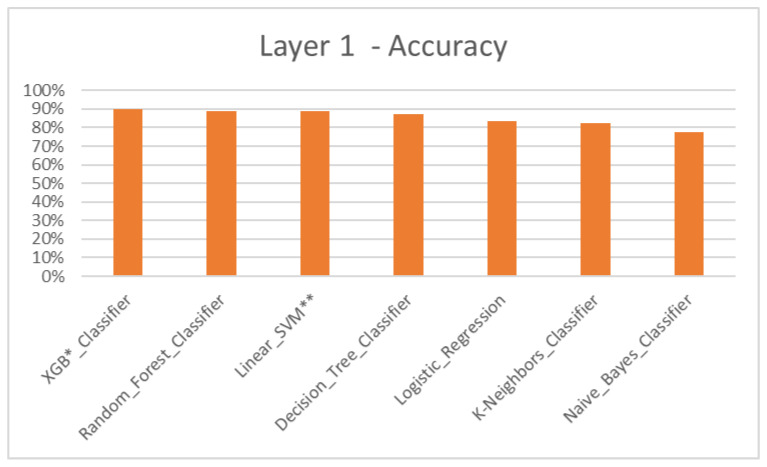
Layer 1 for prediction of skeletal jaw bases I, II and III—Accuracy. * XGB—eXtreme Gradient Boosting. ** SVM—Support Vector Machine.

**Figure 5 dentistry-11-00001-f005:**
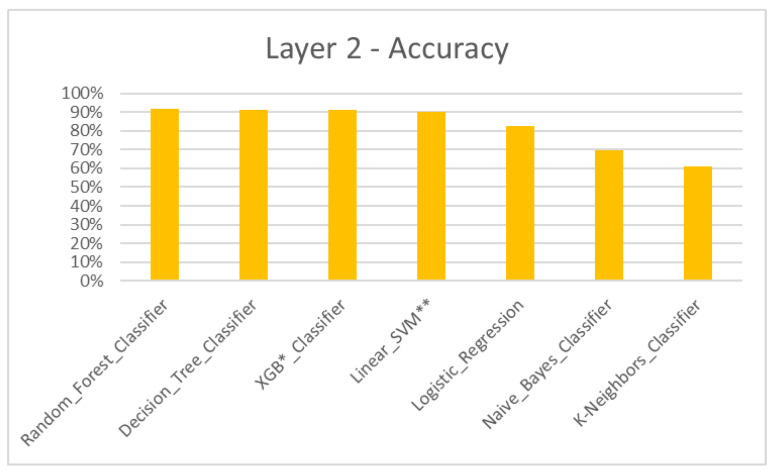
Layer 2 for prediction of extraction and non-extraction treatment for jaw base I—accuracy. * XGB—eXtreme Gradient Boosting. ** SVM—Support Vector Machine.

**Figure 6 dentistry-11-00001-f006:**
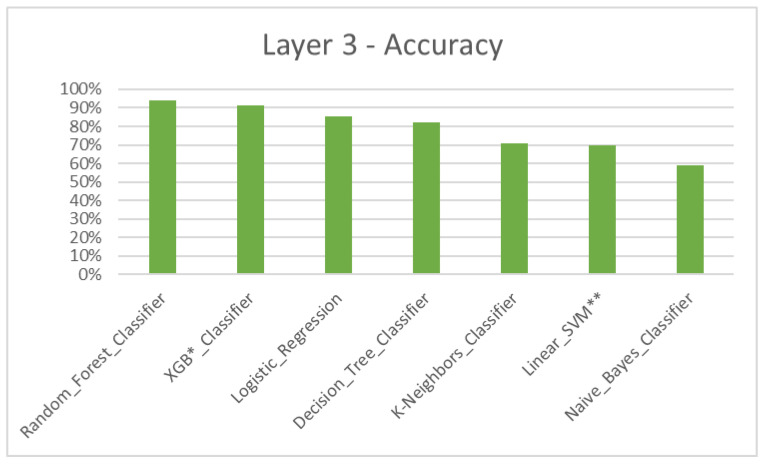
Layer 3 for prediction of growth modulation, camouflage and jaw surgery treatment options for jaw base III—accuracy. * XGB—eXtreme Gradient Boosting. ** SVM—Support Vector Machine.

**Figure 7 dentistry-11-00001-f007:**
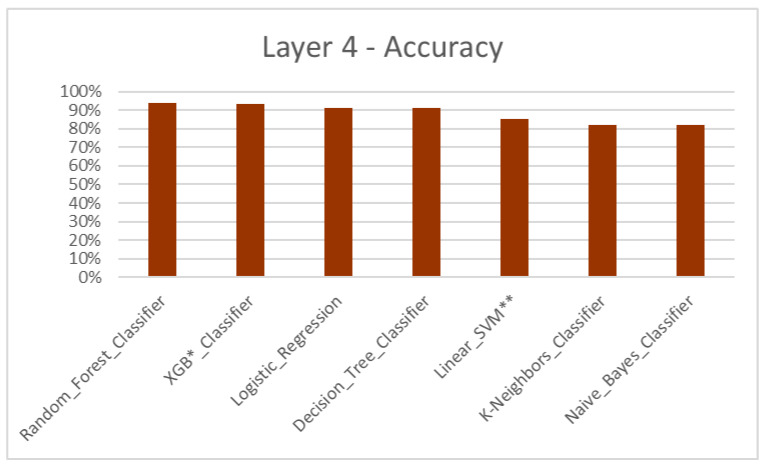
Layer 4 for prediction of growth modulation, camouflage and jaw surgery treatment options for jaw base IV—accuracy. * XGB—eXtreme Gradient Boosting. ** SVM—Support Vector Machine.

**Figure 8 dentistry-11-00001-f008:**
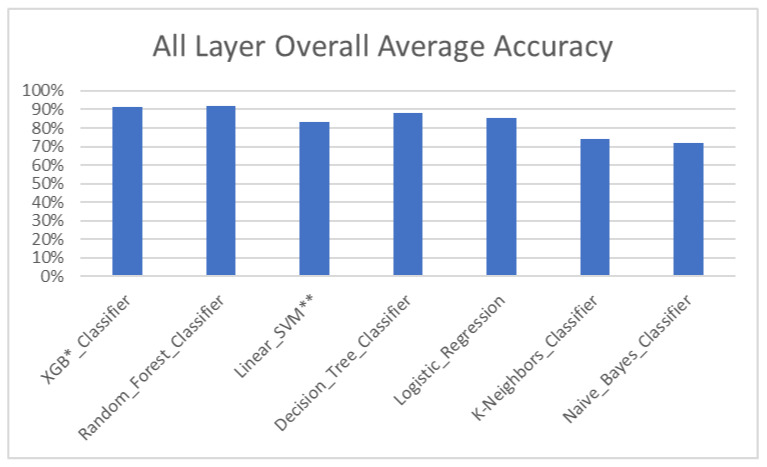
Overall accuracy of the ML model. * XGB—eXtreme Gradient Boosting. ** SVM—Support Vector Machine.

**Figure 9 dentistry-11-00001-f009:**
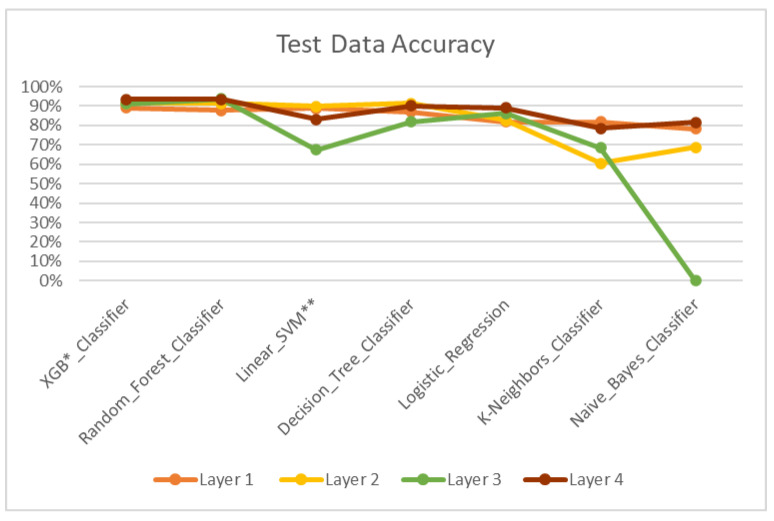
Test data—accuracy. * XGB—eXtreme Gradient Boosting. ** SVM—Support Vector Machine.

**Figure 10 dentistry-11-00001-f010:**
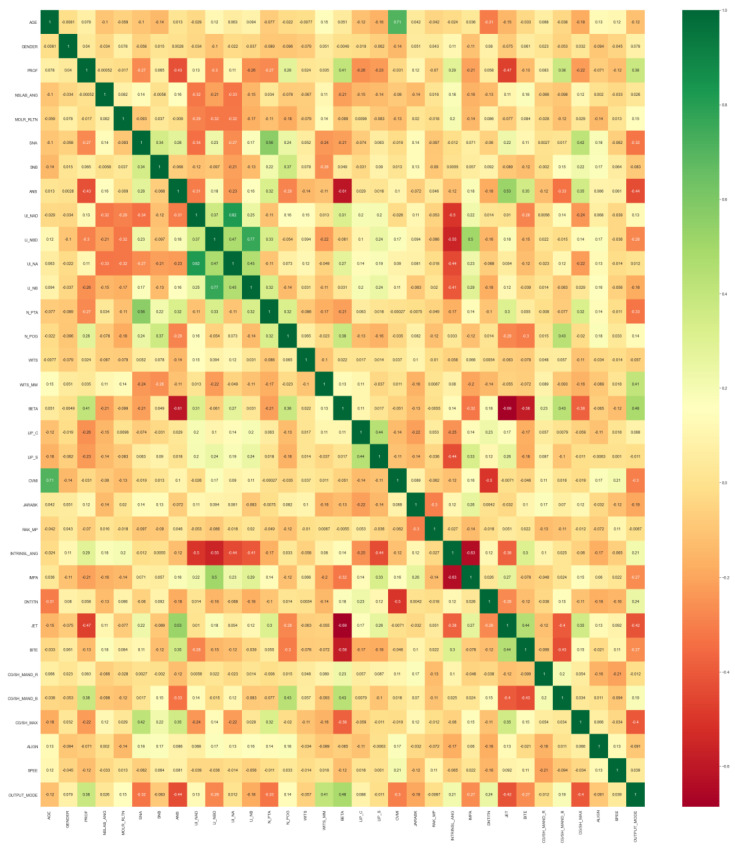
Heat map for correlation between two parameters (greens represent the highest and reds represent the least correlation).

**Figure 11 dentistry-11-00001-f011:**
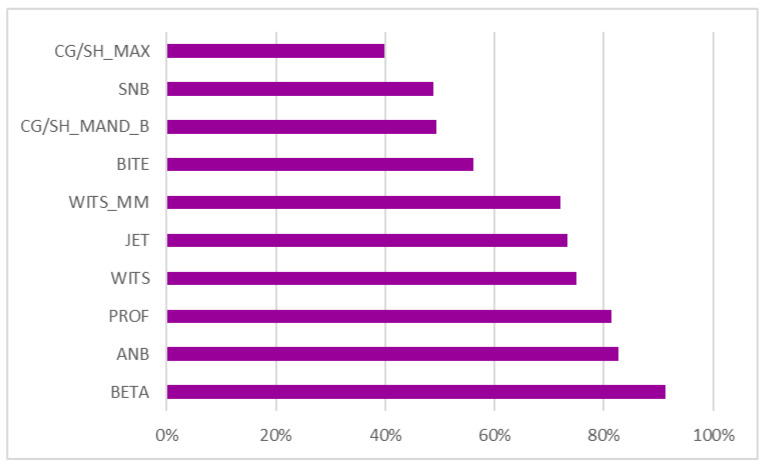
The ten parameters contributing the highest in determining the treatment planning of a case.

**Table 1 dentistry-11-00001-t001:** Input parameters.

Sl No.	Parameter	Sl No.	Parameter	Sl No.	Parameter
1	Patient ID	13	LI–NB mm	25	IMPA^g^
2	Age	14	N perpendicular to Pt. A	26	Dentition
3	Gender	15	N perpendicular to Pog	27	Overjet
4	Profile	16	Wits Appraisal	28	Overbite
5	Nasolabial angle	17	Wits mm	29	Mandibular ramus
6	Molar relation	18	Beta angle	30	Mandibular body
7	SNA ^a^	19	Lip competence	31	Maxilla
8	SNB ^b^	20	Lip strain	32	Alignment
9	ANB ^c^	21	CVMI ^f^	33	Curve of Spee
10	UI–NA ^d^ degrees	22	Jaraback ratio		
11	LI–NB ^e^ degrees	23	Mandibular plane angle		
12	UI–NA mm	24	Interincisal angle		

a—Sella-Nasion-point A angle. b—Sella-Nasion-point B angle. c—point A-Nasion-point B angle. d—upper incisor to Nasion-point A. e—lower incisor to Nasion-point B. f—cervical vertebrae maturation stage. g—incisor mandibular plane angle.

**Table 2 dentistry-11-00001-t002:** Data Dictionary.

Sl No.	Numerical Data	Categorical Data
1	Patient ID	Gender: Male—1 Female—0
2	Age	Profile: Convex—1 Concave—2 Straight—3
3	Nasolabial angle	Molar relation
4	SNA ^a^	Wits Appraisal AO ahead of BO—1 BO ahead of AO—2 AO=BO—3
5	SNB ^b^	Lip competence: Competent—1 Potentially incompetent—2 Incompetent—3
6	ANB ^c^	Dentition Permanent—1 Mixed—2
7	UI–NA ^d^ degrees	Alignment: Aligned—0 Crowding—1 Spacing—2
8	LI–NB ^e^ degrees	
9	UI–NA mm	
10	LI–NB mm	
11	N perpendicular to Pt. A	
12	N perpendicular to Pog	
13	Wits mm	
14	Beta angle	
15	Lip strain	
16	CVMI ^f^	
17	Jaraback ratio	
18	Mandibular plane angle	
19	Interincisal angle	
20	IMPA ^g^	
21	Overjet	
22	Overbite	
23	Mandibular ramus	
24	Mandibular body	
25	Maxilla	
26	Curve of spee	

a—Sella-Nasion-point A angle. b—Sella-Nasion-point B angle. c—point A-Nasion-point B angle. d—upper incisor to Nasion-point A. e—lower incisor to Nasion-point B. f—cervical vertebrae maturation stage. g—incisor mandibular plane angle.

**Table 3 dentistry-11-00001-t003:** Machine Learning algorithm models.

Sl No.	Algorithm
1	Random Forest Classifier
2	XGB * Classifier
3	Logistic Regression
4	Decision Tree Classifier
5	K-Neighbors Classifier
6	Linear SVM **
7	Naïve Bayes Classifier

* XGB—eXtreme Gradient Boosting. ** SVM—Support Vector Machine.

**Table 4 dentistry-11-00001-t004:** Layer 1—Accuracy.

	Model	Accuracy	Precision	Recall	F1
1	XGB *_Classifier	90.00%	88.51%	89.74%	89.00%
2	Random_Forest_Classifier	88.75%	87.50%	87.99%	87.72%
3	Linear_SVM **	88.75%	89.29%	88.95%	88.71%
4	Decision_Tree_Classifier	87.50%	87.27%	87.09%	86.89%
5	Logistic_Regression	83.75%	81.77%	82.19%	81.96%
6	K-Neighbors_Classifier	82.50%	83.84%	82.75%	81.91%
7	Naive_Bayes_Classifier	77.50%	82.48%	80.98%	78.33%
Layer 1 Average		85.54%	85.81%	85.67%	84.93%

* XGB—eXtreme Gradient Boosting. ** SVM—Support Vector Machine.

**Table 5 dentistry-11-00001-t005:** Layer 2—Accuracy.

	Model	Accuracy	Precision	Recall	F1
1	Random_Forest_Classifier	91.60%	92.31%	91.67%	91.29%
2	Decision_Tree_Classifier	91.36%	91.29%	91.29%	91.29%
3	XGB *_Classifier	91.30%	92.31%	91.67%	91.29%
4	Linear_SVM **	90.00%	91.67%	90.00%	89.90%
5	Logistic_Regression	82.61%	82.58%	82.58%	82.58%
6	Naive_Bayes_Classifier	69.57%	74.11%	70.45%	68.62%
7	K-Neighbors_Classifier	60.87%	61.90%	61.36%	60.57%
Layer 2 Average		82.47%	83.74%	82.72%	82.22%

* XGB—eXtreme Gradient Boosting. ** SVM—Support Vector Machine.

**Table 6 dentistry-11-00001-t006:** Layer 3—Accuracy.

	Model	Accuracy	Precision	Recall	F1
1	Random_Forest_Classifier	94.12%	92.67%	95.21%	93.72%
2	XGB *_Classifier	91.18%	90.24%	92.65%	91.21%
3	Logistic_Regression	85.29%	85.24%	87.86%	86.32%
4	Decision_Tree_Classifier	82.35%	82.08%	84.96%	81.98%
5	K-Neighbors_Classifier	70.59%	69.65%	72.39%	68.41%
6	Linear_SVM **	70.00%	67.25%	67.79%	67.43%
7	Naive_Bayes_Classifier	58.82%	43.67%	64.44%	NaN
Layer 3 Average		78.91%	75.83%	80.76%	81.51%

* XGB—eXtreme Gradient Boosting. ** SVM—Support Vector Machine.

**Table 7 dentistry-11-00001-t007:** Layer 4—accuracy.

	Model	Accuracy	Precision	Recall	F1
1	Random_Forest_Classifier	93.60%	93.42%	93.70%	93.36%
2	XGB *_Classifier	93.45%	93.24%	93.44%	93.41%
3	Logistic_Regression	90.91%	92.21%	88.89%	89.18%
4	Decision_Tree_Classifier	90.91%	89.68%	91.11%	90.13%
5	Linear_SVM **	85.00%	83.33%	83.81%	83.22%
6	K-Neighbors_Classifier	81.82%	80.00%	80.00%	78.57%
7	Naive_Bayes_Classifier	81.82%	81.90%	82.22%	81.68%
Layer 4 Average		88.21%	87.68%	87.60%	87.08%

* XGB—eXtreme Gradient Boosting. ** SVM—Support Vector Machine.

**Table 8 dentistry-11-00001-t008:** Overall model accuracy.

	Model	Accuracy	Precision	Recall	F1
1	XGB *_Classifier	91.48%	91.07%	91.88%	91.23%
2	Random_Forest_Classifier	92.02%	91.48%	92.14%	91.52%
3	Linear_SVM **	83.44%	82.88%	82.64%	82.31%
4	Decision_Tree_Classifier	88.03%	87.58%	88.61%	87.57%
5	Logistic_Regression	85.64%	85.45%	85.38%	85.01%
6	K-Neighbors_Classifier	73.94%	73.85%	74.13%	72.37%
7	Naive_Bayes_Classifier	71.93%	70.54%	74.52%	76.21%
Overall Average		83.78%	83.26%	84.19%	83.75%

* XGB—eXtreme Gradient Boosting. ** SVM—Support Vector Machine.

**Table 9 dentistry-11-00001-t009:** Test set accuracy.

	Model	Layer 1	Layer 2	Layer 3	Layer 4
1	XGB *_Classifier	89.00%	91.29%	91.21%	93.41%
2	Random_Forest_Classifier	87.72%	91.29%	93.72%	93.36%
3	Linear_SVM **	88.71%	89.90%	67.43%	83.22%
4	Decision_Tree_Classifier	86.89%	91.29%	81.98%	90.13%
5	Logistic_Regression	81.96%	82.58%	86.32%	89.18%
6	K-Neighbors_Classifier	81.91%	60.57%	68.41%	78.57%
7	Naive_Bayes_Classifier	78.33%	68.62%	NaN	81.68%
	**Average**	**84.93%**	**82.22%**	**81.51%**	**87.08%**

* XGB—eXtreme Gradient Boosting. ** SVM—Support Vector Machine.

## Data Availability

Not applicable.

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
