# Peer review of "Machine Learning Predictive Model as Clinical Decision Support System in Orthodontic Treatment Planning"

_dentistry, 2022, doi:10.3390/dj11010001_

Round 1
Reviewer 1 Report
Introduction and discussion should be improved with the suggested references about tridimensional imaging for diagnosis and treatment planning
The authors can implement their work by describing the importance of correct diagnosis and the tools that can be used to support it, in all areas of dentistry, whether orthodontic, implant, aesthetic or prosthetic, in this sense specify the use of 2D and 3D diagnostic tools, with the references indicated
PubMed ID 34425656
PubMed ID 34425658
PubMed ID 34425663
Materials and methods are well described and pertinent
Results are clearly described and very coherent with the materials and methods
CONCLUSION is correct and interesting.
Author Response
Cover Letter addressing the comments and suggestions
1. The inclusion and exclusion criteria have been put in the text and the respective tables have been removed. Since its a retrospective machine learning study requiring large amounts of data, the inclusion and exclusion criteria have been added to narrow down the data set and make the data specific to the machine learning model being created.
2. All the graphs have been made into figures.
3. The graphs have been automatically generated with the values obtained and cannot be altered. Since the values have up to two decimal points, the Y axis values have decimal points not commas (for e.g., since values are 82.67%, 69.57%, the y axis values are 60.00%,70.00%, 80.00%)
4. Each layer represents different levels in prediction and hence cannot be combined in a single graph. All the layers are represented with different colors.
5. The Heat map (Figure 10) and Feature Importance graph (Figure 11) previously mentioned as graph 7 and 8 cannot be remade. Its automatically generated based on the results. However, the titles have been changed for better understanding and the explanation for the same are in the ‘Discussion’ section of the paper.
6. The research does not contain any references to tridimensional imaging pertaining to orthodontic diagnosis and treatment planning.
7. All the references related to the research are cited in the ‘References’ section.
Reviewer 2 Report
Although this might be interesting manuscript, some changes need to be done.
First, Table 1 and Table 2, inclusion and extrusion criteria are better to put in the text instead in Table. Why authors decided to make those criteria. Please expain.
Graph 1, 2... authors need to change it in the Figures because in the manuscript this is Figure even it present the Graph.
Also in all graphs y axis need to be aligned in all graphs from 0 to 100 %. Please put the numbers without comma ( for example 70 instead of 70.00 ).
I suggest that authors put all layers in one big Graph and every layer colour in different colour.
Figure 9 is not self explain! make the new one and explain what is on x -axis, what on y-axis. Also, change the Title of Figure 7 and Figure 8 to be clearly understand what represent.
Author Response

(The authors gave the same response as above.)

Round 2
Reviewer 2 Report
Thank you for considering most of the suggestions for improving the manuscript. As I emphasized before, all Figures s should be uniform. the y axis must be on everyone from 0 to 100 and the percentages must be written without decimal places. So, 100% instead of 100.00%, 90 5 instead of 90.00% and so on... On Figures that reach less than 100, for example 92 %, correct 92 to 100, and so on...
Author Response
- All the figures have been remade. The Y axis values are uniform throughout with no decimal places and has been corrected accordingly from 0% to 100% in all figures.
- English language and style are improved.